# Scalable Ensemble Diversification
# for OOD Generalization and Detection

## Abstract

Training a diverse ensemble of models has several practical application scenarios, such as model selection for out-of-distribution (OOD) generalization and the detection of OOD samples via Bayesian principles. Previous approaches to diverse ensemble training have relied on the framework of letting the models make the correct predictions for the given in-distribution (ID) data while letting them come up with different hypotheses for the OOD data. As such, they require well-separated ID and OOD datasets to ensure a performant and diverse ensemble and have only been verified in smaller-scale lab environments where such a separation is readily available. In this work, we propose a framework, Scalable Ensemble Diversification (SED), for scaling up existing diversification methods to large-scale datasets and tasks (e.g. ImageNet), where the ID-OOD separation may not be available. SED automatically identifies OOD samples within the large-scale ID dataset on the fly and encourages the ensemble to make diverse hypotheses on them. To make SED more suitable for large-scale applications, we propose an algorithm to speed up the expensive pairwise disagreement computation. We verify the resulting diversification of the ensemble on ImageNet and demonstrate the benefit of diversification on the OOD generalization and OOD detection tasks. In particular, for OOD detection, we propose a novel uncertainty score estimator based on the diversity of ensemble hypotheses, which lets SED surpass all the considered baselines in OOD detection task. Code will be available soon.

## 1 Introduction

Training a diverse ensemble of models is useful in multiple applications. Diverse ensembles are used to enhance out-of-distribution (OOD) generalization, where strong spurious features learned from the in-distribution (ID) training data hinder generalization [30, 31, 28, 23]. By learning multiple hypotheses, the ensemble is given a chance to learn causal features that are otherwise overshadowed by the prominent spurious features [39, 4]. In Bayesian machine learning, diversification of the posterior samples has been studied as a means to improve the precision and efficiency of sample uncertainty estimates [5, 37].

A common strategy to train a diverse ensemble is to introduce two objectives: one for the main task and one for diversification [29, 5, 28, 23]. The main task loss, such as the cross-entropy loss for classification, encourages the hypotheses to solve the task on the labeled ID training set. The diversification loss encourages the hypotheses to diversify the responses on an unlabelled OOD dataset [28, 23] (Figure 1). The datasets for the objectives are separated to avoid contradictory objectives: prediction diversification on the ID set will encourage wrong answers if there is only one correct label.

Submitted to 38th Conference on Neural Information Processing Systems (NeurIPS 2024). Do not distribute.

This strategy, however, requires a separate OOD dataset where the hypotheses may make diverse predictions without harming the main task performance on the ID training samples. Previous work has thus been tested on hypothetical lab settings where the spurious and causal features can easily be controlled to secure separate ID and OOD datasets for diverse ensemble training. It is not clear yet how one could diversify an ensemble of models for realistic, uncontrolled, and large-scale applications (e.g. ImageNet scale) where collecting a separate OOD dataset can be very costly, if not impossible.

To address the scalability challenge, we propose a novel diversification framework, Scalable Ensemble Diversification (SED, Figure 1). We introduce three ingredients. (1) OOD samples are dynamically selected from the ID training samples, on which the models are trained to make different predictions. (2) At each iteration, a subset of model pairs are stochastically selected to construct the disagreement objective, rather than the full list of model pairs. (3) Deep networks are trained to diversify only a few layers at the end, rather than the full networks. This framework allows scaling up existing ensemble diversification methods. In this work, we focus on scaling up the Agree to Disagree (A2D) method [28].

We verify that SED diversifies a model ensemble trained on ImageNet. We demonstrate the benefit of diversification on OOD generalization and OOD

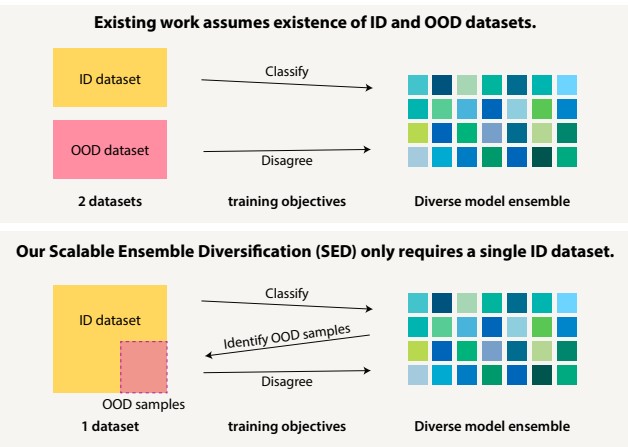

Figure 1: **Existing diversification work vs SED**. Unlike previous diversification approaches that require a separate OOD dataset on which the models are trained to diverge, our Scalable Ensemble Diversification (SED) operates on a single ID dataset where OOD samples are dynamically identified and are used to let the ensemble members diverge.

detection tasks. For the former, we showcase the usage of SED-diversified ensemble in three variants: (a) *vanilla ensemble* of prediction probabilities [22], (b) an average of the model weights through *model soup* [38], and (c) the *oracle selection* of the individual models for each OOD test set [23, 30]. In all three cases, SED achieves a superior generalization to OOD datasets like ImageNet-A/R/C, OpenImages, and iNaturalist.

For OOD detection, we seek multiple ways to use the SED-diversified ensemble: (a) treating them as samples of the Bayesian posterior and (b) using our novel OODness estimate of Predictive Diversity Score (PDS) that measures the diversity of predictions from an ensemble. We show that PDS provides a superior detection of OOD samples like ImageNet-A/R/C, OpenImages, and iNaturalist.

Our contributions are

1. Scalable Ensemble Diversification (SED) framework that scales up existing ensemble methods;

2. Predictive Diversity Score (PDS) that computes the OODness score for samples based on ensemble prediction diversity;

3. First demonstration of the ensemble diversification and its application to OOD generalization and detection at ImageNet level.

The code will be released with the next versions of the manuscript.

## 2 Related work

In this section, we give a short overview of ensembling methods. At first, we speak about ensembles in general and the role of diversity in them (§ 2.1), then we focus on ensembling methods for neural networks and separate them into two big groups. The first group includes algorithms that use loss regularizers (§ 2.2) and the second group covers works that do not modify the training loss (§ 2.3).

## 2.1 Ensembles as a technique

Ensembling is a powerful technique of aggregating the outputs of multiple models to make more accurate predictions and it has been around for decades [12, 21, 18, 2, 3]. It is well known that diversity in ensemble members' outputs leads to better performance of the ensemble compared to the performance of a single model [21] because ensemble members make independent errors [12, 11]. Therefore, one way to reduce DNNs' reliance on spurious correlations is to train multiple models on the same task and make them diverse in terms of errors they make so that their ensemble is less dependent on such correlations.

## 2.2 Neural network ensembles that promote diversity through loss regularizers

Diversity in models can be induced by supplying training loss with a suitable regularizer.

Such regularizers can diversify models' weights [5, 7, 34, 6], features [39, 4], input gradients [29, 30, 31, 33] and outputs [25, 5, 28, 23].

Notably, in [5] authors showed that regularizer of a certain structure that repulses ensemble members' weights or outputs leads to ensembles that provide a better approximation of Bayesian Model Averaging. This idea was later extended by works that repulse ensemble members' features [39] and input gradients [33].

Since the ensemble performs better due to the diversity of errors that ensemble members make [21] we want those members to give pairwise different outputs for the same inputs. Unfortunately, diversity in weights space, input gradient space, or features space does not guarantee such property without additional assumptions due to functional symmetry which means that models can be different in terms of their weights or feature maps and input gradients they produce but still give the same outputs for a given input. That is why we are focused on methods that diversify models' outputs, specifically [28, 23] which are state-of-the-art according to [1] and use regularizer of repulsive nature conceptually similar to [5].

## 2.3 Neural network ensembles that promote diversity without modifying loss

In addition to loss regularizers, there were an uncountable number of different ways to induce diversity in ensembles of neural networks that did not modify the training loss. The most straightforward approach of independently training multiple models of the same architecture by changing only random seeds is called Deep Ensemble [22] which was extended from the Bayesian perspective in [37]. Another solution is to construct an ensemble from models trained with different hyperparameters [36], augmentations [24], or architectures [40]. More computationally efficient direction allows training only one base model inducing diversity by ensembling either checkpoints saved in different local minima along the training trajectory of this base model [19] or models produced by the base model after applying dropout [10] or masking [9] to it. The mixture of experts paradigm can also be viewed as an ensemble diversification technique [41] where diversification happens due to assigning different training samples to different ensemble members.

Despite their conceptual simplicity Deep Ensembles [22] and ensembles of models trained with different hyperparameters [36] are strong baselines for OOD detection [27] and OOD generalization tasks, especially when combined with model souping techniques [38]. That is why we selected them as baselines for our experiments.

# 3 Method

We present our main technical contributions, Scalable Ensemble Diversification (SED, §3.2) and the Predictive Diversity Score (PDS, §3.3).

## 3.1 Preliminaries

We cover background materials before introducing our main technical contributions. We work with a training set $\mathcal{D} := \{x_n, y_n\}_{n=1}^N$, which we refer to as the in-distribution (ID) dataset. For prior diversification methods, we also assume the existence of a separate, unlabeled out-of-distribution

(OOD) dataset $\mathcal{D}^{\mathrm{ood}} := \{x_n^{\mathrm{ood}}\}_{n=1}^{N^{\mathrm{ood}}}$. We write $f(\cdot, \theta)$ for a deep neural network classifier parametrized by $\theta$. $f(x; \theta) \in \mathbb{R}^C$ indicates the logit outputs for $C$ classes for input $x$. We write $p(x) := \mathrm{Softmax}(f(x)) \in [0,1]^C$ for the probability outputs. We consider an ensemble $\{f^1, \cdots, f^M\}$ of $M$ models.

### 3.1.1 Existing ensemble diversification approach

We introduce an existing approach for diversifying an ensemble of models [28, 23]. Two objectives are imposed upon the ensemble of models: the main task loss and the diversification regularization.

For the main task, the community has focused on the classification task. The cross-entropy loss $-\log p_y(x; \theta)$ is used to train the model ensemble $\{f^1, \cdots, f^M\}$ on the ID dataset $\mathcal{D}$:

$$\mathcal{L}_{\mathrm{main}} = \frac{1}{MN} \sum_n \sum_m -\log p_{y_n}^m(x_n; \theta). \tag{1}$$

This encourages each member of the ensemble to behave similarly on the ID dataset.

Different diversification schemes use different diversification regularization loss $\mathcal{L}_{\mathrm{div}}$ applied on pairs $(f^m, f^l)$ of ensemble members. The diversification objective is commonly optimized on the OOD dataset $\mathcal{D}^{\mathrm{ood}}$ to encourage the training of multiple hypotheses on the OOD samples while avoiding clashes with the main task objective. In this work, we focus on the Agree to Disagree [28] method. The diversification loss for a pair $(p^m, p^l)$ is defined as:

$$\mathrm{A2D}(p^m(x), p^l(x)) = -\log\left[p_{\hat{y}}^m(x) \cdot (1 - p_{\hat{y}}^l(x)) + (1 - p_{\hat{y}}^m(x)) \cdot p_{\hat{y}}^l(x)\right] \tag{2}$$

where $\hat{y} := \arg\max_c p_c^m(x)$ is the predicted class for the first model $p^m$. One may symmetrically define $\hat{y}$ to be the prediction for the second model $p^l$; in practice, it does not make a difference [28]. Note that the diversification loss favors $p^l$ to predict a lower likelihood for the prediction by $p^m$, $p_{\hat{y}}^l(x)$, and vice versa. For $M$ models in an ensemble, A2D is applied on the OOD dataset $\mathcal{D}^{\mathrm{ood}}$ for every pair of models $(p^m, p^l)$:

$$\mathcal{L}_{\mathrm{div}} = \frac{1}{N^{\mathrm{ood}} \cdot M(M-1)} \sum_n \sum_{m<l} \mathrm{A2D}(p^m(x_n^{\mathrm{ood}}), p^l(x_n^{\mathrm{ood}})). \tag{3}$$

### 3.2 Scalable Ensemble Diversification (SED)

We present Scalable Ensemble Diversification (SED) that addresses the limitation of the existing ensemble diversification framework that requires a separate OOD dataset. We introduce two main components of SED: dynamic selection of OOD samples within the ID dataset (§3.2.1) and the stochastic selection of pairs to diverge in the optimization iterations (§3.2.2).

### 3.2.1 Dynamic selection of OOD samples

If only the ID training dataset is present, it is difficult to induce diversity in ensemble members, as they are uniformly incentivized to solve the main task objective: given $x$, predict $y$. Hence, previous approaches have introduced a qualitatively disjoint unlabeled set, which we refer to as the OOD dataset, where the ensemble members are encouraged to disagree with each other. The clear separation of ID and OOD datasets for the two objectives matters for ensuring a good balance between the main task performance and the diversity of hypotheses.

Previous works like Pagliardini et al. [28], Lee et al. [23] have performed experiments on small-scale datasets where factors are well-controlled and clean versions of OOD datasets are readily available. Examples include Waterbirds, Camelyon17, CelebA, MultiNLI, C-MNIST, and the Office-Home datasets. For example, for Waterbirds, the ID dataset is set as the cases where the bird's habitat matches with the visual background and the OOD dataset corresponds to the complementary case.

While conceptually desirable, collecting a separate OOD dataset can be highly cumbersome and expensive. For a large-scale dataset like ImageNet, it is highly non-obvious how one could build a corresponding OOD dataset where the underlying feature-label correlations are different from the ID training dataset.

To address this challenge, we consider dynamically identifying an OOD subset of the ID dataset and letting the ensemble diverge on this subset. The desiderata for the identification of OOD samples

within the ID dataset are twofold: (a) we wish to discriminate samples where the ensemble members make mistakes and (b) we only trust the ensemble prediction for the OOD sample identification when the ensemble is sufficiently trained.

We define the sample-wise weight $\alpha_n$ on each ID sample $(x_n, y_n) \in \mathcal{D}$ that satisfy the two conditions:

$$\alpha_n := \frac{\text{CE}(f^1, \cdots, f^M; x_n, y_n)}{\left(\frac{1}{|B|} \sum_{b \in B} \text{CE}(f^1, \cdots, f^M; x_b, y_b)\right)^2} \tag{4}$$

where $\text{CE}(f^1, \cdots, f^M; x_n, y_n) := \text{CE}(\frac{1}{M} \sum_m f^m(x_n), y_n)$ is the loss on the logit-averaged prediction and $B$ is a minibatch that contains the sample $(x_n, y_n)$. $\alpha_n$ is a weight proportional to the ensemble loss on the sample; we thus meet the condition (a). The normalization is designed to handle the condition (b). To see this, consider the batch-wise weight

$$\alpha_B := \frac{1}{|B|} \sum_{b \in B} \alpha_b = \frac{1}{\frac{1}{|B|} \sum_b \text{CE}(f^1, \cdots, f^M; x_b, y_b)}. \tag{5}$$

Note that $\alpha_B$ is now *inversely proportional* to the average cross-entropy loss of the ensemble on the batch $B$. Thus, the overall level of $\alpha_n$ for $n \in B$ is lower for earlier iterations of the ensemble training, where the predictions from the models are not trustworthy yet.

With this definition of sample-wise weight $\alpha_n$ for the diversification objective, we define the SED objective with the A2D loss for the diversification kernel:

$$\mathcal{L}_{\text{SED}} := \mathcal{L}_{\text{main}} + \frac{\lambda}{NM(M-1)} \sum_n \sum_{m<l} \text{stopgrad}(\alpha_n) \cdot \text{A2D}(p^m(x_n), p^l(x_n)), \tag{6}$$

where $\lambda > 0$ controls the overall weight of the diversification term. Note that, compared to Equation 3, this formulation does not rely on the OOD dataset $\mathcal{D}^{\text{ood}}$. Instead, all ID samples are treated as potential OOD samples, where their OODness is softly determined via $\alpha_n$. This enables a seamless adaptation of existing ensemble diversification methods to a relaxed setting where a separate OOD dataset is unavailable.

### 3.2.2 Further tricks for scalability

Ensemble diversification algorithms are often based on pairwise similarities of the members. Pairwise similarity computation scales quadratically with the size of the ensemble $M$. The second term of Equation 6 is an example of this. This is potentially a hurdle when ensemble diversification is to be applied to $M \geq 10$, and the data and parameter sizes are in the order of millions (e.g. ImageNet).

We address this computational challenge by computing the summation of pairwise distances as a stochastic sum. For every minibatch $B$ of SGD iterations, we uniformly-iid sample a subset $\mathcal{I}$ of $\{1, \cdots, M\}$ to compute the diversification term in Equation 6. The procedure is illustrated in the figure on the right.

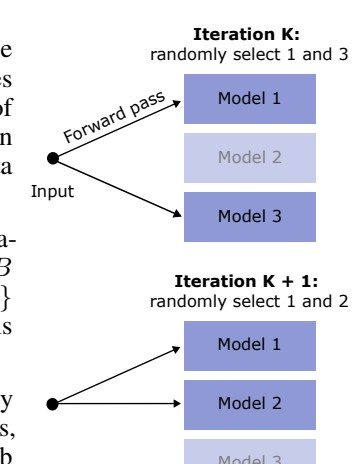

To further speed up the SED training, we consider diversifying only a subset of layers, while freezing the other layers. In our experiments, ensemble members share the same frozen feature extractor of Deit3b [32] pretrained on ImageNet-21k [8] and we diversify only the last two layers of the models.

### 3.3 Predictive Diversity Score (PDS) for OOD Detection

We demonstrate several benefits of the diversified ensembles in §4. One of them is the possibility of using them for detecting OOD samples through the notion of epistemic uncertainty [13]. Given an ensemble of models, a simple baseline for OOD detection is to compute the predictive uncertainty of the Bayesian Model Averaging (BMA) by treating the ensemble members as samples of the posterior $p(\theta|\mathcal{D})$ [22, 37]:

$$\eta_{\text{BMA}} := \max_c \frac{1}{M} \sum_m p_c^m(x). \tag{7}$$

This notion of epistemic uncertainty does not directly exploit the potential diversity in individual models of the ensemble because it averages out the predictions along the model index $m$.

We propose a novel measure for epistemic uncertainty, Predictive Diversity Score (PDS), that directly measures the prediction diversity of the individual members. The formulation is given below:

$$\eta_{\text{PDS}} := \frac{1}{C} \sum_c \max_m p_c^m(x). \tag{8}$$

PDS is a continuous relaxation of the number of unique argmax predictions within an ensemble of models. To see this, consider the special case where $p^m \in \{0, 1\}$ are one-hot vectors. Then, $\max_m p_c^m(x)$ is 1 if any of $m$ predicts $c$ and 0 otherwise. Thus, $\sum_c \max_m p_c^m(x)$ computes the number of classes that at least one of the ensemble members predicts. We show that, with our diverse ensembles, PDS outperforms the DE baseline for the OOD detection task (§4.4).

## 4 Experiments

We verify our contributions, Scalable Ensemble Diversification (SED, §3.2) and Predictive Diversity Score (PDS, §3.3), on ImageNet-scale tasks and datasets. We first verify that SED diversifies the ensemble (§4.2). Then, we demonstrate the application of diversified ensemble to OOD generalization (§4.3) and OOD detection (§4.4) tasks.

### 4.1 Experimental setup

We task the ensemble with the OOD generalization and OOD detection tasks.

**Training settings.** For both tasks, we train an ensemble of models with the SED framework with the A2D [28] diversity regularization using AdamW optimizer [26]. We use the default settings of a batch size of 16, learning rate $10^{-3}$, weight decay 0.01, and the number of epochs 10. The overall diversity weight $\lambda$ is set to 0.1 and the stochastic pairing is done for $|\mathcal{I}| = 2$ models for each SGD batch. We use Deit3b [32] network pretrained on ImageNet21k [8] for all the experiments. Following the speed-up trick in §3.2.2, we use only the last 2 layers of the network. For the in-distribution (ID) dataset where the ensemble is trained to diversify, we use the training split of ImageNet with $|\mathcal{D}| = 1,281,167$. All experiments were ran on RTX2080Ti GPUs with 12GB vRAM and 40GB RAM, each experiment took from 2 to 12 hours depending on the complexity of the training.

**Baselines.** For naive ensemble training, we consider the *deep ensemble* [22] where each ensemble member independently with different random seeds that control the weight initialization and SGD batch shuffling. To match the resource usage of our SED, where we diversify only the last 2 layers of the network, we consider the *shallow ensemble* variant, which is the deep ensemble where only the last 2 layers are trained. We further consider a viable diversification scheme that performs deep ensemble with *varying hyperparameters* [36]. In addition to that, we reimplement A2D [28] and DivDis [23] algorithms and apply them without stochastic model sampling to do classification on labeled samples from ImageNet-Train and disagreement on unlabeled samples from ImageNet-R. For A2D we use frozen feature extractor and a parallel variant of their method which means that all ensemble members are trained simultaneously and not sequentially. The computational complexity of both these approaches scales quadratically with ensemble size which is why they are called Naive A2D and Naive DivDis respectively.

**Evaluation benchmarks.** The generalization performances of the ImageNet-trained ensembles are measured on multiple test datasets, ranging from the in-distribution validation split of ImageNet with 50,000 samples to OOD datasets like ImageNet-A (*A* [17], 7.5k images & 200 classes), ImageNet-R (*A* [16], 30k images, 200 classes), ImageNet-C (*C-i* for corruption strength $i$ [14], 50k images, 1k classes). OpenImages-O (*OI* [35], 17k images, unlabeled), and iNaturalist (*iNat* [20], 10k images, unlabeled). For OOD detection, we task the ensemble with the detection of the above OOD datasets against the ImageNet validation split.

**Evaluation metrics.** For OOD generalization, we use the accuracy. For OOD detection, we use the area under the ROC curve, following [15].

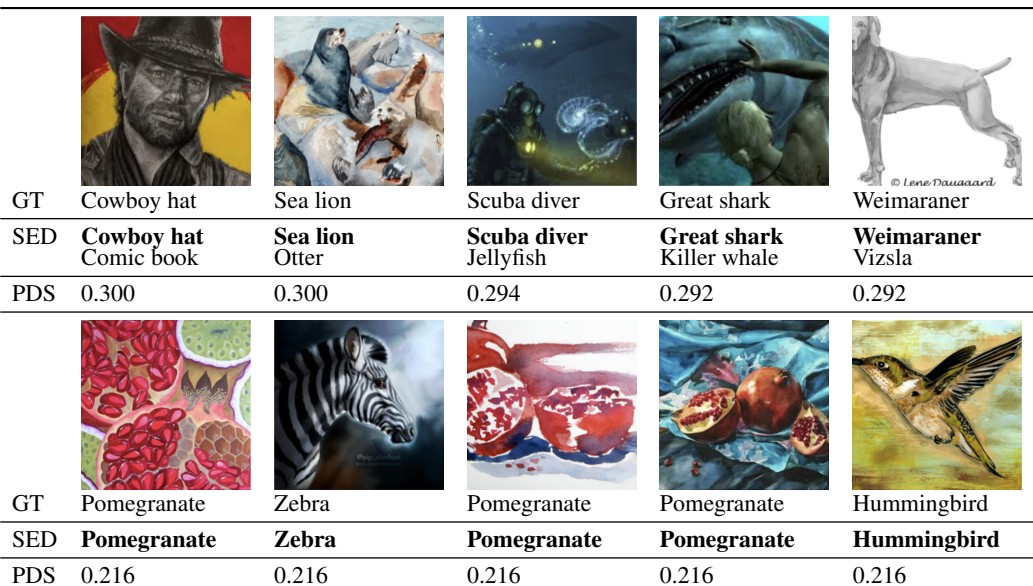

| | | | | |
|---|---|---|---|---|
| GT | Cowboy hat | Sea lion | Scuba diver | Great shark | Weimaraner |
| SED | **Cowboy hat** Comic book | **Sea lion** Otter | **Scuba diver** Jellyfish | **Great shark** Killer whale | **Weimaraner** Vizsla |
| PDS | 0.300 | 0.300 | 0.294 | 0.292 | 0.292 |

| | | | | |
|---|---|---|---|---|
| GT | Pomegranate | Zebra | Pomegranate | Pomegranate | Hummingbird |
| SED | **Pomegranate** | **Zebra** | **Pomegranate** | **Pomegranate** | **Hummingbird** |
| PDS | 0.216 | 0.216 | 0.216 | 0.216 | 0.216 |

Figure 2: **ImageNet-R examples leading to the greatest and least disagreement**. We show the 5 most divergent and 5 least divergent samples according to the SED ensemble. We measure the prediction diversity with the Prediction Diversity Score (PDS) in §3.3. GT refers to the ground truth category. Ensemble predictions are shown in bold, in cases when ensemble members predict classes different from the ensemble prediction we provide them on the next line with standard font.

## 4.2 Diversification

We start with the question of whether Scalable Ensemble Diversification (SED) truly diversify the ensemble at the ImageNet scale. To measure the diversity of the ensemble, we compute the number of unique predictions for each sample for the committee of models (#unique).

Table 1 shows the #unique values for the IN-Val as well as multiple OOD datasets. We observe that the deep ensemble baseline does not increase the diversity dramatically (e.g. 1.09 for C-1) beyond no-diversity values (1.0). Diversification tricks like hyperparameter diversification (1.11 for C-1) or Naive A2D (1.04 for C-1) and DivDis (1.04 for C-1) do not improve the prediction diversity dramatically. On the other hand, our SED increases the prediction diversity across the board (e.g. 5.00 for C-1).

| Method | C-1 | C-5 | iNat | OI |
|---|---|---|---|---|
| Deep ensemble | 1.09 | 1.19 | 1.31 | 1.23 |
| +Diverse hyperparams | 1.11 | 1.32 | 1.48 | 1.33 |
| Naive DivDis | 1.04 | 1.14 | 1.19 | 1.16 |
| Naive A2D | 1.04 | 1.15 | 1.19 | 1.91 |
| SED-A2D | **5.00** | **5.00** | **4.68** | **4.11** |

Table 1: **#unique for ensembles.** We report the #unique on OOD datasets (see §4.1 for the datasets). The ensemble size $M$ is 5 for all methods; it is the max possible #unique value.

Qualitative results on ImageNet-R further verify the ability of SED to diversify the ensemble (Figure 2). As a measure for diversity, we use the Predictive Diversity Score (PDS) in §3.3. We observe that the samples inducing the highest diversity (high PDS scores) are indeed ambiguous: for the first image, where the "cowboy hat" is the ground truth category, we observe that "comic book" is also a valid label for the image style. On the other hand, samples with low PDS exhibit clearer image-to-category relationship.

## 4.3 OOD Generalization

We examine the first application of diversified ensembles: OOD generalization. We hypothesize that the superior diversification ability verified in §4.2 leads to greater OOD generalization due to the consideration of more robust hypotheses that do not rely on obvious spurious correlations.

**Ensemble aggregation for OOD generalization.** As a means to exploit such robust hypotheses, we consider 3 aggregation strategies. (1) *Oracle selection*: the best-performing individual model is chosen from an ensemble [28, 30]. Final prediction is given by $f(x; \theta^{m^\star})$ where

| Method | $M$ | Oracle selection | | | | | Prediction ensemble | | | | | Uniform soup | | | | |
|---|---|---|---|---|---|---|---|---|---|---|---|---|---|---|---|---|
| | | Val | IN-A | IN-R | C-1 | C-5 | Val | IN-A | IN-R | C-1 | C-5 | Val | IN-A | IN-R | C-1 | C-5 |
| Single model | 1 | 85.4 | 37.9 | 44.7 | 75.6 | 38.5 | 85.4 | 37.9 | 44.7 | 75.6 | 38.5 | 85.4 | 37.9 | 44.7 | 75.6 | 38.5 |
| Deep ensemble | 5 | **85.4** | 37.9 | 44.9 | 75.7 | 38.6 | **85.4** | 39.9 | 46.3 | 75.7 | 38.6 | **85.3** | 36.7 | 44.6 | 75.5 | 38.3 |
| +Diverse HPs | 5 | **85.4** | **38.5** | **45.4** | 77.4 | **40.7** | **85.4** | 39.9 | 46.5 | 76.0 | 39.0 | **85.3** | 35.3 | 44.1 | 75.9 | 38.7 |
| Naive DivDis | 5 | 85.2 | 35.8 | 40.8 | 77.2 | 40.2 | 85.1 | 36.3 | 41.8 | 77.2 | 40.2 | 84.8 | **40.7** | 42.5 | 76.2 | 38.9 |
| Naive A2D | 5 | 85.2 | 36.6 | 44.3 | 77.3 | 40.4 | 85.1 | 37.8 | 45.2 | 77.2 | 40.3 | 84.5 | 39.3 | 45.1 | 75.5 | 39.1 |
| SED-A2D | 5 | 85.1 | 38.3 | 45.3 | 77.2 | 40.4 | 85.3 | **42.4** | **48.1** | **77.3** | **40.6** | **85.3** | 40.3 | **46.1** | **77.3** | **40.6** |
| Deep ensemble | 50 | **85.5** | 38.1 | 45.2 | 75.7 | 38.6 | **85.5** | 38.8 | 45.8 | 75.6 | 38.5 | **85.4** | 37.5 | 45.0 | 75.5 | 38.4 |
| +Diverse HPs | 50 | 85.5 | 38.5 | 45.6 | **77.5** | **40.8** | 85.5 | 42.5 | 48.5 | **76.0** | 39.0 | **85.4** | 36.4 | 44.8 | **75.9** | 38.8 |
| SED-A2D | 50 | 82.6 | **39.0** | **45.8** | 74.4 | 38.3 | 83.5 | **50.9** | **54.4** | 75.8 | **39.3** | 83.5 | **39.2** | **46.5** | 75.8 | **39.3** |

Table 2: **OOD generalization of ensembles.** Models are trained on the ImageNet training split. $M$ is the ensemble size. For Naive DivDis and A2D, we use the ImageNet-R as the OOD datasets where the respective diversification objectives are applied.

$m^\star := \arg\max_m \mathrm{Acc}(f^m, \mathcal{D}^{\mathrm{ood}})$. (2) *Prediction ensemble* is a vanilla prediction ensemble where the logit values are averaged: $\frac{1}{M}\sum_m f^m(x)$ [38]. (3) *Uniform soup* [38] averages the weights themselves. Final prediction is given by $f(x; \frac{1}{M}\sum_m \theta^m)$.

**SED improves OOD generalization for ensembles.** We show the OOD generalization performances of ensembles in Table 2, for the three ensemble prediction aggregation strategies described above. We observe that our SED framework (SED-A2D) results in superior OOD generalization performances for all three strategies. SED-A2D is particularly strong in prediction ensemble (e.g. 48.1% for $M = 5$ and 54.4% for $M = 50$ on ImageNet-R) and uniform soup (e.g. 46.1% for $M = 5$ and 46.5% for $M = 50$ on ImageNet-R). We contend that the increased ensemble diversity contributes to the improvements in OOD generalization. We also remark that the SED framework (SED-A2D) envelops the performance of Naive A2D in this ImageNet-scale experiment. Together with the superiority of computational efficiency (as discussed at the end of § 4.4) of SED-A2D over the Naive A2D, this demonstrates that SED fulfills its purpose of scaling up ensemble diversification methods like A2D.

**Deep ensemble is a strong baseline.** We also note that deep ensemble, particularly with diverse hyperparameters, provides a strong baseline, outperforming dedicated diversification methodologies under the oracle selection strategy when $M = 5$. It also provides a good balance between ID (ImageNet validation split) and OOD generalization.

## 4.4 OOD Detection

We study the impact of ensemble diversification on OOD detection capabilities of an ensemble. Once an ensemble is trained, we compute the epistemic uncertainty, or likelihood of the sample being OOD, following two schemes, $\eta_{\mathrm{BMA}}$ and $\eta_{\mathrm{PDS}}$ introduced in §3.3.

**SED and PDS together lead to superior OOD detection performances.** We show the OOD detection results in Table 3. For the BMA scores, deep ensemble remains a strong baseline. In particular, when the hyperparameters are varied ("+Diverse HPs"), the detection AUROC reaches the maximal performances among the ensembles using the BMA scores. The quality of PDS is more sensitive to the ensemble diversity, as seen in the jump from the deep ensemble (e.g. 0.589 for OI) to the diverse-HP variant (0.889). However, when the ensemble

| Method | $\eta$ | C-1 | C-5 | iNat | OI |
|---|---|---|---|---|---|
| Single model | BMA | 0.615 | 0.833 | 0.958 | 0.909 |
| Deep Ensemble | BMA | 0.619 | 0.835 | 0.958 | 0.911 |
| +Diverse HPs | BMA | **0.642** | **0.861** | **0.969** | **0.923** |
| Naive DivDis | BMA | 0.598 | 0.843 | 0.966 | 0.922 |
| Naive A2D | BMA | 0.594 | 0.835 | 0.966 | 0.916 |
| SED-A2D | BMA | 0.641 | 0.845 | 0.960 | 0.915 |
| Deep Ensemble | PDS | 0.565 | 0.625 | 0.592 | 0.589 |
| +Diverse HPs | PDS | 0.643 | 0.849 | 0.926 | 0.889 |
| Naive DivDis | PDS | 0.600 | 0.851 | 0.969 | 0.939 |
| Naive A2D | PDS | 0.599 | 0.850 | 0.971 | 0.939 |
| SED-A2D | PDS | **0.686** | **0.896** | **0.977** | **0.941** |

Table 3: **OOD detection via ensembles.** For each OOD dataset (C-1, C-5, iNat, and OI), the ensembles are tasked to detect the respective OOD samples among ID samples (ImageNet validation split). We show the AUROC scores for the OOD detection task. Ensemble size is fixed at $M = 5$. $\eta$ refers to the epistemic uncertainty computation framework discussed in §3.3.

is sufficiently diverse, such as when trained
with SED-A2D, the PDS leads to high-quality OODness scores. SED-A2D with PDS achieves the
best AUROC across the board, including the BMA variants.

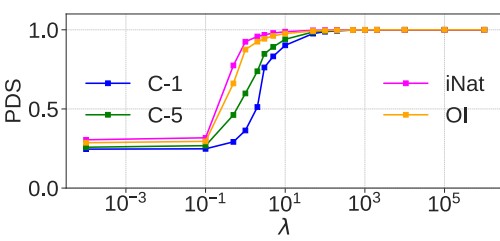 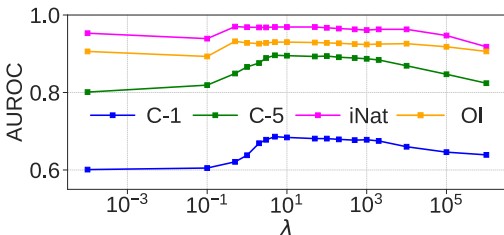

Figure 3: **Impact of diversity regulariser on OOD detection**. We show the model answer diversity, measured by PDS, and the OOD detection performance, measured by AUROC, against $\lambda$ values, the loss weight for the disagreement regularizer term.

**Impact of diversification parameter $\lambda$.** We further study the impact of ensemble diversification on the OOD detection with the PDS estimator. In Figure 3, we observe that strengthening the diversification objective (higher $\lambda$) indeed leads to greater diversity (higher PDS), with a jump at around $\lambda \in [10^{-1}, 10^1]$. This range corresponds to the jump in the OOD detection performance (higher AUROC).

**Influence of ensemble size.** How ensemble size influences performance of our method? We can see that increasing ensemble size helps to improve AUROC for OOD detection on C-1 (Figure 4). Increasing ensemble size marginally helps, but using 5 models provides already a significant improvement over the smallest possible ensemble of size 2. It is also important to mention, that SED framework is computationally more efficient w.r.t. ensemble size $M$ than Naive

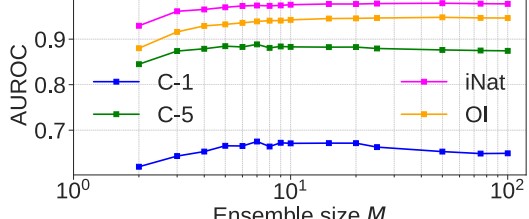

Figure 4: **Impact of ensemble size on OOD detection**.

A2D and Naive DivDis: since we train ensembles for the fixed number of epochs, training complexity for SED is $O(1)$ thanks to stochastic model pairs selection, while for Naive A2D and Naive DivDis it is $O(M^2)$.

# 5    Conclusion

Ensemble diversification has many implications for treating one of the ultimate goals of machine learning, handling out-of-distribution (OOD) samples. By training a large number of plausible hypotheses on an in-distribution (ID) dataset, an OOD-generalizable hypothesis may appear. Moreover, the diversity of hypotheses lets us distinguish ID samples from OOD samples by measuring the degree of divergence in ensemble members' predictions. Despite conceptual benefits, diverse-ensemble training has previously remained a lab-bound concept for several reasons. First, previous approaches required a separate OOD dataset that may nurture diverse hypotheses. Second, computational complexities of previous pairwise diversification objectives increase quadratically with the ensemble size.

We have addressed the challenges through the novel Scalable Ensemble Diversification (SED) framework. SED identifies the OOD-like samples from a single dataset, bypassing the need to prepare a separate OOD dataset. SED also employs a stochastic pair selection algorithm which reduces the quadratic complexity of previous approaches to a constant cost per SGD iteration. We have demonstrated good performances by SED on the OOD generalization and detection tasks, both at the ImageNet scale, a largely underexplored regime in the ensemble diversification community. In particular, for OOD detection, our novel diversity measure of Predictive Diversity Score (PDS) amplifies the benefits of diverse ensembles for OOD detection. The code to reproduce the results of our experiments will provided with the next revision of the manuscript.

**Limitations**

We do not provide theoretical justification for the method. Our experiments were conducted on models with a frozen feature extractor.

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
