# OpenReview forum: "Scalable Ensemble Diversification for OOD Generalization and Detection"
_NeurIPS.cc/2024/Conference — Submitted to NeurIPS 2024_

### Official Review · Reviewer_UGvs · 2024-07-08

**Soundness:** 2
**Presentation:** 2
**Contribution:** 2
**Rating:** 5
**Confidence:** 3

**Summary:**

The paper presents SED, a method for scaling up existing diversification methods to large-scale datasets and tasks. SED identifies the OOD-like samples from a single dataset, bypassing the need to prepare a separate OOD dataset. Experimental results demonstrated good performances by SED on the OOD generalization and detection tasks.

**Strengths:**

1. SED scales up existing ensemble methods.
2. According to the author's experimental results, SED demonstrates its application to OOD generalization and detection at ImageNet level.
3. Simple method, and easy to understand.

**Weaknesses:**

1. I am very confused about the dataset division for OOD detection task in the paper. The distribution should refer to “label distribution” in OOD detection [1], which means that OOD samples should not have overlapping labels w.r.t. training data. In the paper, the ID dataset is ImageNet-1K, while the OOD dataset for the OOD detection task includes ImageNet-C (Table 3). Their label spaces overlap, which is clearly incorrect. I don't believe the experiments conducted in this paper fall under the category of OOD detection. I suggest the authors refer to relevant literature on OOD detection, such as OpenOOD [1].

2. The ablation studies are insufficient. For instance, the number of layers being diversified is a hyperparameter. I believe conducting ablation experiments on this would make the paper more solid.

3. I think the experiments in the paper are not comprehensive enough. For example, how does it perform on small-scale datasets? Although it may not be fair to compare with methods using real OOD datasets, this could provide insights into SED's performance from multiple perspectives.

4. The comparative methods in the paper are not comprehensive enough. How does it perform compared to existing OOD generalization and OOD detection methods? If SED is complementary to existing methods, how much improvement can it bring?

5. The paper claims to speed up pairwise divergence computation, but no results are shown. Could authors demonstrate specifically how much speedup was achieved?

6. typos:
6.1 Line 61: "We verify that SEDdiversifies a model..."
6.2 Line 68: "In all three cases, SEDachieves a superior generalization..."

[1] Yang et al, OpenOOD: Benchmarking Generalized Out-of-Distribution Detection, IJCV 2024.

**Questions:**

1. Where is the batch-wise weight $\alpha_B$ used in the method?

**Limitations:**

I think the author should discuss more about the limitations of the method proposed in the article, such as the computational time required for the method proposed in the article compared with other methods.

---

> ### Author Rebuttal · Authors · 2024-08-07
>
> We thank the reviewer for the thorough review and positive comments about the method/results. The questions/comments are very useful for improving the paper. We added a **number of new results** (in the attached PDF) and made **numerous clarifications** to the paper (summarized below).
>
> ---
> **W1: Overlap of label spaces.**
>
> We will clarify upfront in the paper that **this work considers two types of shifts - semantic and covariate [a - j]**. Openimages-O and iNaturalist represent datasets with semantic shifts and their labels sets are disjoint with ImageNet1k label set. ImageNet-C dataset is a representative example of covariate shift for OOD detection. Its labels coincide with labels of ImageNet-1K, while the "style" of the images does not. Note that this dataset was also used in the latest version of the OpenOOD [a] referenced by the reviewer.
>
> ---
> **W2: Additional ablations.**
>
> We **performed additional ablations as requested** (different number of layers for the DeiT-3b feature extractor). See Table 5 in the attached PDF. Happy to add others if deemed necessary.
>
> ---
> **W3: Performance on small-scale datasets?**
>
> **The whole point of this paper is to scale up** the work presented in the original A2D and DivDis papers to a more realistic and useful ~ImageNet level.
>
> Nonetheless, we performed additional experiments on the (small) Waterbirds dataset (see Table C in the authors rebuttal), since both A2D and DivDis also provided results on it. We report the worst group (test) accuracy for ensembles of size 4. We did not use the stochastic sum for our method to factor out its influence. A2D and DivDis use the validation set for disagreement. While DivDis discovers a better single model, the ensemble is clearly better with the proposed SED method.
>
>
> ---
> **W4: Additional comparisons**
>
> Thanks for the suggestion. We performed **additional comparisons** (see Table A in the author rebuttal). The proposed SED with model-soup aggregation performs better than the compared methods [m,n] across all OOD datasets.
>
> ---
> **W5: Demonstration of speedup.**
> We performed an additional evaluation (Table 7 in the attached PDF) that shows the benefit of controlling the "subset size" in the stochastic sum ($\mathcal{I}$) on the speed of training an ensemble. For example, to train an ensemble of size 5, the time required for 1 epoch **grows from 53s to 585s without stochastic sum** ($\mathcal{I}=2$). We could not train an ensemble of 50 models without stochastic sum with our resources, but it already requires 7244s for ($\mathcal{I}=10$) vs 2189s for ($\mathcal{I}=2$).
>
> ---
> **Q1: Where is $\alpha_B$ used?**
>
> We do not use it directly in our method. It was provided in text to highlight the fact that weights $\alpha_n$ are small when ensemble is undertrained because the sum of all weights in batch equals $\alpha_B$ and is inversely proportional to average classification loss on this batch.
>
> ---
> **Limitations.** Thanks for the suggestion. We added an evaluation of the computational time of the proposed method vs. others (see the attached PDF).

---

> > ### Comment · Reviewer_UGvs · 2024-08-11
> >
> > The reviewer thanks the authors for the response.
> >
> > Regarding weakness 4, the authors' added baselines are all OOD generalization methods. I suggest adding more OOD detection baselines to make the article more complete.

---

> ### Author Response · Authors · 2024-08-11
>
> Thanks for the suggestion. We are keen to include additional comparisons in the final version to strengthen the paper. Suggestions of specific methods are welcome.
>
> We provided results for additional OOD detection methods in the authors' rebuttal (see Table B: Ensemble Entropy, Average Entropy, Mutual Information, Average Energy, and the A2D disagreement as an uncertainty score [k,p]).
>
> The comparison with BMA/deep ensembles (already in the paper) is probably the most important since this is considered state-of-the-art in recent studies [s, v, w].
>
> Summary of results: the proposed PDS proves superior to all compared methods on all covariate and semantic shift datasets, i.e. C-1, C-5, iNaturalist, OpenImages.
>
> [k] Pagliardini et al., *Agree to disagree: Diversity through disagreement for better transferability*, ICLR 2022.
>
> [p] Xia and Bouganis, *On the Usefulness of Deep Ensemble Diversity for Out-of-Distribution Detection*, ICLR 2022.
>
> [s] Ovadia et al., *Can you trust your model's uncertainty? evaluating predictive uncertainty under dataset shift*, NeurIPS 2019.
>
> [v] Mukhoti et al. *Deep deterministic uncertainty: A new simple baseline*, CVPR 2023.
>
> [w] Dusenberry et al. *Efficient and scalable bayesian neural nets with rank-1 factors.* International conference on machine learning, PMLR 2020.

---

> > ### Author Response · Authors · 2024-08-13
> >
> > We thank the reviewer for the extended discussion. Since we only have around 24 hours for the discussion period, we would like to ask the reviewer to share any final requests for us to respond in time. Please also reconsider the original scores, if the concerns are adequately addressed.

---

> > > ### Comment · Reviewer_UGvs · 2024-08-13
> > >
> > > Dear authors,
> > >
> > > Thank you for the response.
> > >
> > > Most of the issues have been addressed. I have raised the score.
> > >
> > > Best,
> > > Review UGvs

---

> > > > ### Author Response · Authors · 2024-08-13
> > > >
> > > > We thank the reviewer for the acknowledgement and the score update. However, since the score is still 4, a borderline reject, we would like to understand if there are any remaining issues that need to be addressed. We would be happy to provide any further clarification, elaboration, or even experiments within the next 24 hours.

---

> > > > > ### Comment · Reviewer_UGvs · 2024-08-13
> > > > >
> > > > > Dear authors
> > > > >
> > > > > I think adding some post-hoc methods to the comparative approaches for OOD detection tasks would make the paper more comprehensive. Methods like Energy [1] and MaxLogit [2] could be good additions.
> > > > >
> > > > > More important, I’m confused about the setting of detecting both semantic and covariate shift simultaneously. For covariate shift samples, our goal should be to improve classification accuracy. For semantic shift samples, our goal should be to have the model reject classification to avoid unpredictable behavior. Therefore, I don’t quite understand how, after detecting both semantic and covariate shift samples, we can determine which samples belong to covariate shift and need further classification by the model.
> > > > >
> > > > > If the above concerns can be resolved, I am willing to raise my rating.
> > > > >
> > > > > [1] W Liu, et al. Energy-based Out-of-distribution Detection. In NeurIPS, 2020.
> > > > >
> > > > > [2] D Hendrycks, et al. Scaling Out-of-Distribution Detection for Real-World Settings. In ICML, 2022.
> > > > >
> > > > > Best, Review UGvs

---

> > > > > > ### Author Response · Authors · 2024-08-13
> > > > > >
> > > > > > Thank you for your response, we present below the requested results and clarifications.
> > > > > >
> > > > > > ## Additional post-hoc OOD detection methods
> > > > > >
> > > > > > We have added the results for Energy [1] and Max Logit [2] in Table D below.
> > > > > >
> > > > > > Table D - extended OOD detection baselines
> > > > > >
> > > > > > |                    | When added    | C-1   | C-5   | iNat      | OI        |
> > > > > > | ------------------ | --- | --------- | --------- | --------- | --------- |
> > > > > > | BMA                |  Manuscript   | 0.641     | 0.845     | 0.960     | 0.915     |
> > > > > > | PDS                | Manuscript    | **0.686** | **0.896** | **0.977** | **0.941** |
> > > > > > | a2d_score          | Author's rebuttal    | 0.685     | **0.896** | 0.962     | 0.917     |
> > > > > > | Average Entropy    | Author's rebuttal    | 0.580     | 0.825     | 0.960     | 0.916     |
> > > > > > | Ens. Entropy       | Author's rebuttal    | 0.580     | 0.826     | 0.960     | 0.916     |
> > > > > > | Mutual Information | Author's rebuttal    | 0.503     | 0.539     | 0.586     | 0.576     |
> > > > > > | Max Logit          | Current comment    |     0.673      |   0.874       |   0.809        |    0.829       |
> > > > > > | Energy     | Current comment     | 0.633     | 0.858     | **0.977** | 0.908     |
> > > > > >
> > > > > > ## For covariate shift, goal is to improve classification accuracy. For semantic shift, goal is to have the model reject classification.
> > > > > >
> > > > > > Thank you for the further clarification of the concern; we understand the viewpoint. However, fundamentally, we do not agree that the subsequent action for covariate-shift samples should always be to improve classification accuracy. The goal of abstained prediction is to reject samples where the model is *likely to be wrong*, regardless of whether the cause of that is due to covariate shift or semantic shift [a].
> > > > > >
> > > > > > In fact, covariate shifts are often cited as the motivation for abstained prediction. [b] studies the abstained prediction problem from a theoretical point of view where the OOD samples are covariate-shifted samples (§2.2 and throughout the paper). [c] studies the problem from an empirical perspective where the OOD samples are sourced from CIFAR-C (Figure 11) or ImageNet-C (Figure 7), examples of covariate-shifted samples of the respective ID datasets.
> > > > > >
> > > > > > In line with this observation, we point to many OOD detection benchmarks that use the covariate shift as the source of OODness. Authors in [d] have measured the sensitivity of epistemic uncertainties against covariate shifts on MNIST like rotation and translation (Figure 1) as well as the shifts on CIFAR and ImageNet defined by the CIFAR-C and ImageNet-C datasets (Figure 2). Mukhoti et al. [e] have used CIFAR vs CIFAR-C (Figure 1) and ImageNet vs ImageNet-O (Table 2) as the OOD detection benchmarks.
> > > > > >
> > > > > > We argue that the detection of covariate-shifted samples as an instance of OOD detection task is not only well-motivated but also a widely-adopted practice in the OOD detection community.
> > > > > >
> > > > > > [a] Learning with Rejection. ALT 2016.
> > > > > >
> > > > > > [b] Towards optimally abstaining from prediction with OOD test examples. NeurIPS 2021.
> > > > > >
> > > > > > [c] Uncertainty Estimates of Predictions via a General Bias-Variance Decomposition. AISTATS 2023.
> > > > > >
> > > > > > [d] Can You Trust Your Model's Uncertainty? Evaluating Predictive Uncertainty Under Dataset Shift. NeurIPS 2019.
> > > > > >
> > > > > > [e] Deep Deterministic Uncertainty: A New Simple Baseline. CVPR 2023.
> > > > > >
> > > > > > ## How, after detecting both semantic and covariate shift samples, can we determine which samples are covariate shift and need further classification by the model.
> > > > > >
> > > > > > Following our previous argument, covariate-shifted samples do not always need to be subsequently classified by the model. In many applications, it benefits to still reject these samples.
> > > > > >
> > > > > > Independent of this, the separation of covariate shift vs semantic shift cases is an interesting topic on its own. However, this is beyond the scope of the paper.

---

### Official Review · Reviewer_nXDD · 2024-07-10

**Soundness:** 3
**Presentation:** 3
**Contribution:** 2
**Rating:** 5
**Confidence:** 3

**Summary:**

This paper presents a new method which directly encourages ensemble diversification on selected ID datapoints without the need for a separate OOD dataset. They also introduce a new measure of epistemic uncertainty which measures the diversity of the final predictions of each model, and suggest a speedup of comparing pairwise disagreement via random sampling.

**Strengths:**

- The paper is well-written, and the presentation of the method is easy to understand.
- The experiments cover a wide range of OOD datasets.
- The methods are intuitive and can be inexpensively applied to existing ensemble diversification algorithms.
- SED-A2D outperforms other baselines when using uniform soup or prediction ensembles for OOD generalization, and also achieves the highest AUROC for OOD detection.

**Weaknesses:**

- It appears that utilizing this new training objective leads to a loss in ID accuracy, since it encourages members of the ensembles to diverge. This tradeoff between ID accuracy and OOD accuracy may not be desirable in many settings. Overall, the paper emphasizes the improved OOD performance but does not show its impact on ID data for many experiments, such as the ablation studies for OOD detection, model diversity, etc.
- The stochastic computation of pairwise disagreement seems incremental, and there is no work comparing this stochastic implementation with the traditional expensive one. It would be helpful to include an ablation study to understand the accuracy vs performance tradeoff.
- There are many other methods for OOD detection beyond MSP with BMA (eg [1]). How PDS does compare against other baselines?
- Deep ensembles remain competitive in many settings, and the best values for C-1 and C-5 OOD generalization are still achieved using ensembles.

[1] Xia and Bouganis: On the Usefulness of Deep Ensemble Diversity for Out-of-Distribution Detection https://arxiv.org/abs/2207.07517

**Questions:**

- The selection of OOD samples does not appear to be tied to a specific method. Have you tried applying this method to other like DivDis?
- It's interesting that diversifying on ID samples improves OOD generalization compared to diversifying on ImageNet-R (Table 2). What was the setup for these experiments? Did the ensembles see fewer "OOD" datapoints in the latter?
- In Table 1, I'm surprised that SED-A2D has each model giving a different prediction for C-1, but OOD detection AUROC using PDS is quite low. Does this imply that SED-A2D also has very high disagreement on ID data?

**Limitations:**

- The approach sacrifices ID accuracy for OOD generalization/detection.
- The experiments only showed the result of finetuning the last two layers.

---

> ### Author Rebuttal · Authors · 2024-08-07
>
> We thank the reviewer for the thorough review and positive comments about the intuitiveness of the method, the extensiveness of the evaluation, and the empirical results. The questions/comments are very useful for improving the paper. We added a **number of new results** (in the attached PDF) and made **numerous clarifications** to the paper (summarized below).
>
> ---
> **W1: Tradeoff between ID and OOD performance.**
>
> We supplemented the results tables with the ID performance were missing (see Table 1 and 4 in the attached PDF). The tradeoff in ID vs. OOD performance is inherent to the very nature of the problem and has been discussed extensively in the literature on domain generalization [u] (e.g. a model discarding spurious features because they do not generalize OOD necessarily becomes less performant on domains where these features are informative). The advantage of the "diversification" approach is that one member of the ensemble with particularly high ID performance can be selected, if this is the objective/selection criterion.
>
> ---
> **W2: Ablation study on stochastic sum.**
>
> We performed additional experiments on the subset size $\mathcal{I}$ of the stochastic sum, see Table 7 in the attached PDF. They evaluate the speed up from different $\mathcal{I}$ vs. performance. As we can see by comparing SED with $I=2$ and $I=5$ which have 42.6 vs 37.6 drop on IN-A and 48.1 vs 44.9 drop on IN-R speed-up helps OOD generalization. However, results for OOD detection are not so straightforward - having growth of AUROC on covariate shift datasets, e.g.  0.896 vs 0.903 on C-5 while dropping from 0.977 to 0.970 on iNat for semantic shift datasets.
>
> ---
> **W3: Comparison of PDS against other baselines.**
>
> We performed additional comparisons, see Table B in the authors rebuttal (ensemble entropy, average entropy, mutual information, and average energy [p]). BMA/deep ensemble is state-of-the-art as reported in recent studies [s]. Happy to include other methods if deemed necessary by the reviewer.
>
> ---
> **W4: Deep ensembles remain competitive in many settings, and the best values for C-1 and C-5 OOD generalization are still achieved using ensembles.**
>
> Deep ensembles are indeed competitive, especially with diverse sets of hyperparameters across its members. However, our SED-A2D shows better OOD generalization on 15/23 of the cases  (i.e. 65%) in Table 1.
>
> ---
> **Q1: Application to DivDis.**
>
> Our preliminary experiments with DivDis showed that it was unable to create a diverse ensemble of >2 models. Intrinsic limitations of DivDis were indeed reported in prior work (see Appendix Section F.7 of A2D).
>
> ---
> **Q2: It's interesting that diversifying on ID samples improves OOD generalization compared to diversifying on ImageNet-R (Table 2). What was the setup for these experiments? Did the ensembles see fewer "OOD" data points in the latter?**
>
> SED makes the soft selection of "OOD" datapoints via $\alpha$ term in Equation 6. It is difficult to directly compare the number of used "OOD" datapoints for SED and when IN-R is used.
>
>
> One possible explanation for why diversifying on IN-R did not help is that many samples in IN-R are still within the distribution of the ID dataset (IN-train). Therefore, enforcing disagreement on them may severely hinder the training of the main task for the ensemble, leading to a decrease in OOD accuracy. It is better to let the adaptive loss $\alpha$ decide which samples are safe for disagreeing - i.e. the ones which already have high CE loss.
>
> ---
> **Q3: In Table 1, I'm surprised that SED-A2D has each model giving a different prediction for C-1, but OOD detection AUROC using PDS is quite low. Does this imply that SED-A2D also has very high disagreement on ID data?**
>
> Yes, SED-A2D also has quite high disagreement on ID data. We show #unique and PDS on ID and OOD datasets, ID accuracy and OOD detection scores for the models in Table 1, 4 in the attached PDF.
>
> It is not a problem because what matters is the relative values of PDS between ID and OOD samples. The PDS values (in parentheses) are respectively 4.16 and 3.98 for the covariate shift detector and 3.53 and 1.54 for the semantic shift detector (see Table 1 in the attached PDF). This enables OOD detection at a state-of-the-art level.

---

> ### Comment · Reviewer_nXDD · 2024-08-12
>
> Thank you for providing so many additional experiments and ablations during the rebuttal process! Based on these new results, I am willing to increase my score to a 5.

---

> > ### Author Response · Authors · 2024-08-12
> >
> > We thank the reviewer for appreciating our rebuttal and raising their score.

---

### Official Review · Reviewer_SDvr · 2024-07-11

**Soundness:** 2
**Presentation:** 2
**Contribution:** 2
**Rating:** 5
**Confidence:** 4

**Summary:**

The paper aims to train a diverse ensemble of models via a framework called Scalable Ensemble Diversification. This framework does not require an additional dataset of OOD inputs, as it identifies OOD samples from a given ID dataset. It then encourages the ensemble to return diverse predictions (disagreement) on these OOD samples. Furthermore, the framework makes use of stochastic summation to speed up the disagreement computation. Results are shown for different tasks like generalisation, OOD detection on different OOD datasets.

**Strengths:**

- The high level idea of removing the need for a separate OOD dataset and speeding up the diversification computation can be useful in practice.
- The writing is clear and easy to follow.

**Weaknesses:**

1. It is not clear why the method works, additional ablations studies would be useful.
    - Naive A2D (and DivDis) uses IN-R data to compute the disagreement loss, which could give the method an advantage as it has access to the OOD data. However, it has a lower accuracy on IN-R. There seems to be two methodological differences between A2D and SED-A2D, the OOD data and use of stochastic sum. Given that A2D computes the full pairwise disagreement and stochastic sum is meant to reduce cost rather than improve performance, why does SED-A2D perform better? It would have been useful to compare two methods that only differs on the OOD data used. E.g., SED-A2D without the stochastic sum.
    - From eqn 6, it looks like the two terms have contradicting objectives. For a “OOD” point, the first term encourages all models to classify the point correctly, but the second term encourages models to have different predictions on the same point. These objectives can be challenging to balance.
2. The writing clearly explains the method or setup, but sometimes stops short of giving further insights. For example,
    - Further analysis of experimental results
        - Table 2 why does having more ensemble component (5→50) make the SED-A2D results worse? Similar trends can also be seen in Tab 4 for C-1 or C-5. Could it be because the stochastic sum does not scale with more models?
        - Why was #unique used in Tab 1 to measure diversity when the Predictive Diversity Score was just introduced?
        - Why does oracle selection perform worse compared to simple average in Tab 2? I would expect otherwise given that there is privilege information.
    - Components of the method can be better motivated
        - Why was the A2D loss chosen instead of other losses e.g. DivDis?
        - Why is optimizing Eqn 6 preferable to e.g., forming an OOD dataset from the ID data based on the errors of DeiT or even from the errors from an ensemble of models, similar to imagnet-a, and using existing techniques like [23,28].
    - “collecting a separate OOD dataset can be very costly, if not impossible”.
        - There are cheap ways to introduce OOD samples to an ID dataset, e.g., simple augmentations/transformations to the input. Why are these methods not preferable?
4. One of the main contributions involves speeding up the disagreement computation. There does not seem to be experimental details or results on this. E.g., a subset of models is chosen, what is the size of this subset? How does performance for generalization/detection change with and without this speedup?

**Questions:**

1. It would be interesting to see if using eval datasets other than IN-R as OOD data results in similar trends in Tab 2. It seems like deep ensemble and its variant and SED-A2D, i.e., the methods that do not make use of external OOD data, tend to perform better.
2. Instead of computing the detection scores, how does the proposed diversity measure correlate with correct predictions?
3. In table 1, what is the IN-Val #unique?

**Limitations:**

Yes the limitations were adequately addressed.

---

> ### Author Rebuttal · Authors · 2024-08-07
>
> We thank the reviewer for the thorough review and positive comments about the idea/clarity. The questions/comments are very useful for improving the paper. We added a **number of new results** (in the attached PDF) and made **numerous clarifications** to the paper (summarized below).
>
> ---
> **W1: Ablations; why the method works.**
>
> - *Compare two methods that only differ on the OOD data used.*
> See the additional experiment in Table 7 in the attached PDF. The ablation shows SED-A2D without the stochastic sum. Now we can compare the line of SED with $I=5$ in Table 7 in the attached PDF and Naive A2D from Table 2, 3 in the manuscript as they both do not include stochastic sum. Namely, in OOD generalization they become on par with each other with 44.9 vs 44.3 on IN-R dataset and 37.6 vs 37.8 on IN-A dataset, while in OOD detection SED is still superior in covariate shift case achieving 0.903 vs 0.850 AUROC on C-5 and on par in semantic shift case achieving 0.941 vs 0.939 on OI. As expected, average time spent on training for one epoch significantly decreases for ensembles of various sizes, e.g. from 585s to 53s.
> - *Why does stochastic sum improve performance?*
> Stochastic sum is similar to bagging [t]: each model is trained on a different subset of the training data at each epoch. That imposes further diversity in addition to the disagreement loss. This is why it is unsurprising that the stochastic sum improves OOD generalization.
> - *Loss terms with contradicting objectives.*
> The $\alpha_n$ (eq. 4) indeed balances the opposing effects of these two terms, depending on the "OODness" of the sample. Its value is proportional to the sample-wise classification loss. For an OOD sample, $\alpha_n$ is far greater (e.g. in the order of ~15 vs 0.01), which dwarfs the effect of the cross-entropy loss.
>
> ---
> **W2: Additional insights.**
>
> Thanks for these suggestions! We added a discussion of these points to the paper, which makes the analysis of the results much more informative.
>
> - *Why does oracle selection perform worse compared to simple average in Table 2? (despite using privileged information)*
> Results with "oracle selection" refer to the selection of the best model **per test dataset**, not **per test sample**. It is possible that the best individual member is still worse than an ensemble or averaged weight of the members on a test dataset.
>
> - *Why was the A2D loss chosen instead of other losses e.g. DivDis?*
> Our preliminary experiments with DivDis showed that it was unable to create a diverse ensemble of >2 models. Intrinsinc limitations of DivDis were indeed reported in prior work (see Appendix Section F.7 of [k]).
>
> - *Tables 2,4: why more ensemble components make SED-A2D worse?*
> This is a mere artefact of the experimental conditions, which are now made clearer in the paper. Computational constraints dictated us to fix the number of epochs to a smaller value, such that each individual model is undertrained and suboptimal as seen in the table.
>
> - *Why is Eq. 6 preferable to an OOD dataset from model errors?*
> The proposed approach is more straightforward and efficient, since there is no need to train an initial model to determine OOD samples. We also performed additional experiments showing on-par or better performance by SED (see Table 6 in the attached PDF). Namely, in OOD generalization SED approach (called joint in the table) tied with 2-staged one on IN-A while being slightly worse on IN-R - 48.1 vs 48.5. However, in OOD detection SED approach leads to superior performance across all OOD datasets having 0.896 vs 0.845 AUROC on covariate shift dataset C-5 and 0.941 vs 0.911 on semantic shift dataset OI.
>
> - *Can we introduce OOD samples via augmentations?*
> This is an interesting suggestion. We performed a preliminary evaluation of this idea (see "synth" in Table 8 in the attached PDF). We performed ensemble diversification with small random crops (size sampled within 8-100%) on 30k samples of the ImageNet training set (same size as IN-R). That resulted in achieveing better OOD detection performance across the board in combination with any regularizer - having 0.647 vs 0.600 AUROC on C-1 and 0.974 vs 0.971 AUROC on iNat.
>
> ---
> **W3: What is the model batch size for the stochastic sum? How does performance for generalization/detection change with and without this speedup?**
>
> The size selected for each batch is $\mathcal{I}=2$ (L237). We report additional experiments in Table 7 in the attached PDF. They evaluate the speed up from different $\mathcal{I}$ vs. performance. As we can see by comparing SED with $I=2$ and $I=5$ which have 42.6 vs 37.6 drop on IN-A and 48.1 vs 44.9 drop on IN-R speed-up helps OOD generalization. However, results for OOD detection are not so straightforward - having growth of AUROC on covariate shift datasets, e.g. 0.896 vs 0.903 on C-5 while dropping from 0.977 to 0.970 on iNat for semantic shift datasets.
>
> ---
> **Q1. Other datasets than IN-R for the OOD samples.**
>
> We performed additional experiments with ImageNet-A ("Natural Adversarial Examples"). See Table 8 in the attached PDF. They show that disagreement on IN-A and IN-R have almost identical results on both OOD generalization and OOD detection tasks when using A2D regularizer. With the biggest gap in accuracy on IN-R for Div regularizer at 45.2 for disagreeing on IN-A vs 41.8 for disagreeing on IN-R. Similar drop can be observed on IN-A: 37.8 vs 36.3.
>
>
>
> ---
> **Q2: Instead of computing the detection scores, how does the proposed diversity measure correlate with correct predictions?**
>
> We provide an additional analysis in Figure-Table 3 in the attached PDF. It visualizes the ensemble accuracy vs. PDS for IN-Val, C-1 and C-5. Based on the figure, the accuracy seems to correlate negatively with PDS.
>
> ---
> **Q3: In Table 1, what is the IN-Val #unique?** We added this and additional details to the table (see the attached PDF, Table 1).
>
> ---

---

> > ### Author Response · Authors · 2024-08-13
> >
> > Thank you for the review. Since we only have around 24 hours for the discussion period, we would like to kindly ask the reviewer to tell whether they acknowledge our rebuttal and share any additional requests for us to respond in time. Please also reconsider the original scores, if the concerns are adequately addressed.

---

> > > ### Comment · Reviewer_SDvr · 2024-08-14
> > >
> > > Thanks for the clarifications and additional experiments, they will be useful additions to the paper. I will raise my score.

---

> > > > ### Author Response · Authors · 2024-08-14
> > > >
> > > > We thank the reviewer for raising their score and appreciating our clarifications with additional experiments.

---

### Official Review · Reviewer_CkrP · 2024-07-12

**Soundness:** 2
**Presentation:** 3
**Contribution:** 2
**Rating:** 5
**Confidence:** 3

**Summary:**

Ensembles of diverse models have shown promising signs for out-of-distribution (OOD) generalization.
To boost diversity, some methods require a set of OOD examples for measuring the disagreement among models.
The desired OOD examples, however, can be difficult to obtain in practice.
This paper proposes to dynamically draw OOD samples from the training data during training.
This is done by assigning a higher OOD score to examples with a greater loss in each mini-batch.
To make the diversification process across multiple models more efficient, the authors propose a stochastic approach that only diversifies a small sample of models at each iteration.
The resulting diversified models give rise to the notion of a diversity score for uncertainty estimation and can be used for OOD detection.

**Strengths:**

- The paper introduces several reasonable improvements to a state-of-the-art method, A2D, making it more scalable and practically feasible.
- The empirical performance looks good. It is a bit surprising that the proposed method can outperform A2D which has access to “true” OOD datasets.

**Weaknesses:**

- The notion of “OOD samples” in an ID dataset is confusing. The actual implementation, i.e. assigning a higher “OOD-ness” weight to training examples with a greater loss, is more like identifying “hard” training examples rather than just OOD samples. Calling them “OOD samples” somewhat obfuscate their nature. They are not arbitrary OOD samples but hard samples within the support of the ID dataset. It is not obvious why diversifying models’ predictions on such samples would help. Is such prediction diversification always conducive to OOD generalization? If not, when would the proposed method work or break? These relevant theoretical questions are not answered satisfactorily in the current manuscript.
- The connection between SED and PDS is weak; PDS is not well justified. The A2D diversification loss can also be seen as a measure for prediction diversity, like PDS. Why choose PDS instead for OOD detection? Furthermore, is PDS really a good measure for epistemic uncertainty? Imagine two cases. In the first case, two models confidently (with probability 1) predict the same class for an input example, while in the second case, the two models assign uniform probability to all classes for another example. The PDS for these two examples are exactly the same, yet the models are much less confident (or more uncertain) in the second case. Meanwhile, BMA does not have this issue.
- The baselines are relatively limited. There are many other diversification methods which do not require a separate OOD dataset [1, 2, 3]. How does the proposed method compare with these methods? Can the authors also comment on why BMS is the only considered baseline for OOD detection?
- The definition of #unique values is not very clear. Table 1 shows SED-A2D has extremely large #unique values. On C-1 dataset, the value is 5, the maximum possible value. If my understanding is correct, does this suggest that all the 5 models disagree with each other on every C-1 example? If so, this suggests that for many examples, 4 out of 5 models are probably wrong. Why is this more of a good sign than a bad one?

[1] Rame, Alexandre, et al. "Diverse weight averaging for out-of-distribution generalization." Advances in Neural Information Processing Systems 35 (2022): 10821-10836.
[2] Chu, Xu, et al. "Dna: Domain generalization with diversified neural averaging." International conference on machine learning. PMLR, 2022.
[3] Lin, Yong, et al. "Spurious feature diversification improves out-of-distribution generalization." arXiv preprint arXiv:2309.17230 (2023).

**Questions:**

Please see the weaknesses.

**Limitations:**

The authors only briefly mentioned two limitations of the work. I don't notice any potential negative societal impact.

---

> ### Author Rebuttal · Authors · 2024-08-07
>
> Thanks to the reviewer for the thorough review and encouraging comments. The questions and suggestions are very helpful to improve the paper, and we propose **several improvements** (details below) that should make the final version much clearer.
>
> We also added a number of **new results** (in the attached PDF) that were requested by other reviewers and clearly strengthen the paper.
>
> ---
> **W1: Terminology ("OOD samples" in an ID dataset)**
>
> Thanks for pointing this out, the choice of words was indeed confusing. The paper now simply uses "hard (ID) samples".
>
> ---
> **W1b: Theoretical questions.**
>
> Indeed these are important questions. A formal theoretical treatment is out of the scope of this paper but prior work has given support to the idea of diversification ([l]], [k], [r] on underspecification). We propose to summarize these points in the paper as summarized below.
>
> - *Why would diversifying models' predictions on hard samples help?*
> - *Is such prediction diversification always conducive to OOD generalization?*
>
> The difficulty of OOD generalisation is rooted in the task being underspecified, i.e. multiple hypotheses are consistent with the training data. Diversification methods allow discovering a set of such hypotheses. In the proposed method, the assumption is that enforcing diversity in prediction space on *hard* samples can drive this set of hypotheses to contain one with better generalization properties. In particular, the chosen "hard samples" are assumed to be those where the model could make several valid predictions while still generalizing correctly on other "easier" samples.
>
> The validity of these assumptions depends on a complex interaction between the data and the inductive biases of the model. Its formal study is an important direction for future investigations.
>
> ---
> **W2: Connection between SED and PDS.**
>
> The comments by the reviewer are very relevant. We propose to clarify these points with an additional discussion in the paper summarized below.
>
> *Does PDS measure epistemic uncertainty?* Epistemic uncertainty measures how "abnormal" a data point is w.r.t. the training distribution. We propose to measure this degree of abnormality a given sample through the diversity of model predictions on it. Let us consider the following example [borrowed from the reviewer]. The predictions from an ensembles of two members on two samples $v$ and $w$:
> - $p_1(v)=[0,1]$ and $p_2(v)=[1,0]$
> - $p_1(w)=[1/2,1/2]$ and $p_2(w)=[1/2,1/2]$
>
> Even though the naive ensemble returns the same prediction $(p_1+p_2)/2=[1/2,1/2]$, we would evaluate $v$ as more likely to be OOD w.r.t. the training distribution, as the predictions by ensemble members are more diverse. BMA would fail to capture this effect unlike PDS.
>
> ---
> **W3: Baselines.**
>
>
> Thanks for the suggestion. We have added a number of **additional comparisons** to the paper.
>
> - OOD generalization: see Table A in the authors rebuttal. The proposed SED with model-soup aggregation performs better than the compared methods [m,n] across all OOD datasets. We could not obtain the code for [o] (which pursues the different goal of *feature*-space diversification).
> - OOD detection: see Table B in the authors rebuttal. BMA/deep ensemble is state-of-the-art in several recent studies [s]. We also added comparisons with standard baselines: Ensemble Entropy, Average Entropy, Mutual Information, Average Energy [p].
>
> We're happy to include other methods if deemed necessary by the reviewer.
>
> ---
> **W4: Definition of #unique.**
>
> Thanks for pointing this out. We clarified the definition in the paper.
>
> - *Does the value 5 means that all 5 models disagree with each other on every C-1 example?* Yes.
> - *This suggests that for many examples, 4 out of 5 models are probably wrong. Why is this a good sign?*
> If the models are to be ensembled, this extreme diversity could hinder generalisation. If the goal is to perform OOD detection (PDS score), the extreme diversification on OOD samples is beneficial as the degree of diversification is informative about whether a sample is OOD or not.
>
> ---

---

> > ### Comment · Reviewer_CkrP · 2024-08-08
> >
> > Thank you for the clarifications and for providing the additional experiment results.
> >
> > About PDS, I was referring to the following example:
> > - $p_1(v) = p_2(v) = [1,0]$
> > - $p_1(w) = p_2(w) = [1/2, 1/2]$
> >
> > In this case, PDS does not seem to capture the uncertainty in the prediction of $w$ since the PDS of $v$ and $w$ is the same. I see why PDS would do better than BMA in certain cases, but perhaps PDS also has some non-trivial limitations (or assumptions). I don't think they are adequately discussed. It is still not very clear to me why PDS are generally better than BMA as shown by the experiments.

---

> ### Author Response · Authors · 2024-08-10
>
> Thank you for the clarification. We show below how the provided example does not invalidate PDS as a metric of epistemic uncertainty. We also clarify our understanding of epistemic uncertainty and the soundness of ensembles in measuring it. The final version of the paper should be clearly improved by including this discussion.
>
> **Definition of epistemic uncertainty**
>
> Epistemic uncertainty is defined by a lack of training data in certain input regions [a,b].
>
> **Ensembles and epistemic uncertainty**
>
> The lack of supervision in OOD regions means that multiple models exist with similar training risk. They agree on training/ID data but disagree in their predictions on OOD examples [c,d,e]. This is the reason why the rate of agreement on OOD data is a valid measure of epistemic uncertainty.
>
> **Example given by the reviewer**
>
> Regardless of how much entropy each ensemble member exhibits, what eventually matters for the epistemic uncertainty is the degree of agreement among the members. It makes sense that both $v$ and $w$ have the identically low epistemic uncertainty as long as $p_1(v)=p_2(v)$ and $p_1(w)=p_2(w)$. The overall entropy of the predictions matters more for predictive uncertainty (concerned with the correctness of predictions) and aleatoric uncertainty (data uncertainty stemming from the entropy of the ground-truth $p(y|x))$. Epistemic uncertainty is thus related to the ensemble agreement rather than entropies [c,d,e].
>
> We prepared a concrete demonstration of case suggested by the reviewer [1/2, 1/2], [1/2, 1/2] to be added to the appendix. We use a combined dataset of ImageNet-Val (ID) and OpenImages-O (OOD). We compute the predictive distribution of our SED ensemble on this data. We retain examples with a highly-similar distribution across ensemble members (i.e. high agreement) then rank them by decreasing entropy of the predictive distribution. As we expected/argued, the top examples (high entropy) do correspond to ID samples (from ImageNet-Val). Entropy thus cannot be used to reliably separate ID from OOD samples.
>
> **Soundness of PDS**
>
> PDS is precisely measuring the agreement among ensemble members. Unlike BMA, PDS disregards the overall entropy and focuses purely on the agreement. Previous studies [c,d,e] and our empirical results (Table 3 in the paper) support that PDS is a more suitable measure to determine the *OODness* of a sample than BMA and entropy.

---

> > ### Comment · Reviewer_CkrP · 2024-08-12
> >
> > Thank you for the further clarifications. My concerns have mostly been addressed, and thus I have raised my score to borderline accept.

---

> > > ### Author Response · Authors · 2024-08-12
> > >
> > > We thank the reviewer for the discussion about the role of PDS in measuring epistemic uncertainty and for appreciating our clarifications.

---

### Official Review · Reviewer_mQsd · 2024-07-13

**Soundness:** 2
**Presentation:** 3
**Contribution:** 3
**Rating:** 5
**Confidence:** 3

**Summary:**

The paper proposes Scalable Ensemble Diversification (SED) to extend existing diversification methods to large-scale datasets and tasks where ID-OOD separation may not be possible, and also propose Predictive Diversity Score (PDS) as a novel measure for epistemic uncertainty. Extensive analysis and experiments support the effectiveness of the proposed modules.

**Strengths:**

The logic of this paper is very clear, the motivation is reasonable, and the proposed method has been proven to be effective in analysis and experiments. The figures and tables in the paper are also relatively clear.

**Weaknesses:**

1. Although the experiments are diverse, I am not sure if the comparison is comprehensive. Can more explanation and discussion be added?

2. The feature extractor used is frozen. Is the proposed method robust enough to different feature extractors? What will the performance be if the feature extractor is also involved in the training?

**Questions:**

The proposed framework is interesting, but can it potentially be extended to improve a wide range of OOD tasks?

**Limitations:**

The authors have addressed the limitations.

---

> ### Author Rebuttal · Authors · 2024-08-07
>
> Thanks to the reviewer for recognizing the key values of the paper - clear motivation, clear writing, and experimental results supporting the effectiveness of the method. The questions are also very useful to further improve the paper as detailed below.
>
> We also added a number of **new results** (in the attached PDF) that were requested by other reviewers and clearly strengthen the paper.
>
> ---
> **W1: Additional comparisons**
>
> Thanks for the suggestion. We performed **additional comparisons** (see Table A in the authors rebuttal). The proposed SED with model-soup aggregation performs better than the compared methods [m,n] across all OOD datasets, namely, the respective accuracies gaps are: 38.1 vs 36.7 on IN_A, 45.2 vs 44.4 on IN_R, 77.3 vs 76.0 on C-1 and 40.6 vs 39.1 on C-5.
>
> ---
> **W2: Other/non-frozen feature extractor.**
>
> Indeed the method is applicable to other feature extractors. We performed **additional experiments** (Table 5 in the attached PDF). These results while almost identical for OOD generalizatoin show improvements on OOD detection with ResNet-18 (the main paper used DeiT-3b). Namely, AUROC grows from 0.670 to 0.686 on C-1 dataset with covariate shift and from 0.802 to 0.812 on OI dataset with semantic shift.
>
> We are also preparing additional results with non-frozen feature extractor. They should be ready during the discussion phase and we will be added to the final version of the paper
>
> ---
> **W3: Applicability to other OOD tasks?**
>
> Yes indeed. We did not want to overclaim the applicability of the method but we propose to add the following discussion to the paper. We currently focus on OOD generalization and OOD detection, yet other relevant settings include:
> - *Domain adaptation*: after diversifying the ensemble on the source domain, the ensemble member most suited to the target domain could be selected using labeled samples from the target domain.
> - *Unsupervised domain adaptation/test-time training*: similarly, the ensemble member most suited the target domain could be selected unsing an unsupervised objective (e.g. [q]) from the UDA/TTT literature.
> - *Continual learning*: diversifying the ensemble on the inital domain could provide multiple hypotheses to eliminate/adapt as new domains come in. Alternatively, further diversification could be triggered by training novel members that disagree on new domains to facilitate generalization to these new domains.

---

> > ### Comment · Reviewer_mQsd · 2024-08-09
> >
> > I don't have more problems and I am willing to keep my score.

---

> ### Author Response · Authors · 2024-08-09
>
> We thank the reviewer for the response to the rebuttal. We assume that there are some remaining weaknesses of the paper that do not allow to increase the score. Could you please share them as well to improve the final manuscript?

---

### Author Rebuttal · Authors · 2024-08-07

We thank the reviewers for recognising the value of our work and for providing constructive feedback to improve the manuscript. Along with the individual rebuttal, we provide here supporting tables, figures, and references in the markdown text and the PDF file. They are referenced as "Table X in the author rebuttal" or "Table X in the attached PDF".

Table A - Baselines comparison
|      | IN_VAL | IN_A | IN_R | C-1  | C-5  |
|------|--------|------|------|------|------|
| SED  |   85.2 | **38.1** | **45.2** | **77.3** | **40.6** |
| DIWA |   **85.3** | 35.3 | 44.1 | 75.9 | 38.7 |
| DNA  |   84.4 | 36.7 | 44.4 | 76.0 | 39.1 |

Table B - Uncertainty scores comparison
|                    | C-1 det | C-5 det | iNat  | OI    |
|--------------------|---------|---------|-------|-------|
| BMA                |   0.641 |   0.845 | 0.960 | 0.915 |
| PDS                |   **0.686** |   **0.896** | **0.977** | **0.941** |
| a2d_score          |   0.685 |   **0.896** | 0.962 | 0.917 |
| Average Energy     |   0.633 |   0.858 | **0.977** | 0.908 |
| Average Entropy    |   0.580 |   0.825 | 0.960 | 0.916 |
| Average Max Prob   |   0.673 |   0.874 | 0.809 | 0.829 |
| Ens. Entropy       |   0.580 |   0.826 | 0.960 | 0.916 |
| Mutual information |   0.503 |   0.539 | 0.586 | 0.576 |

Table C - Waterbirds
|  | Single | Ensemble |
| -------- | ------ | -------- |
| ERM      |  76.5      |     72.0     |
| DivDis   |    **87.2**    |      78.3    |
| a2d      |   78.3     |      78.3    |
| SED      | 83.5       |     **80.6**     |


[a] Zhang, Jingyang, et al. "Openood v1. 5: Enhanced benchmark for out-of-distribution detection." arXiv preprint arXiv:2306.09301 (2023).

[b] Hendrycks, Dan, and Thomas Dietterich. "Benchmarking neural network robustness to common corruptions and perturbations." arXiv preprint arXiv:1903.12261 (2019).

[c] Hendrycks, Dan, et al. "The many faces of robustness: A critical analysis of out-of-distribution generalization." Proceedings of the IEEE/CVF international conference on computer vision. 2021.

[d] Recht, Benjamin, et al. "Do imagenet classifiers generalize to imagenet?." International conference on machine learning. PMLR, 2019

[e] Yang, William, Byron Zhang, and Olga Russakovsky. "ImageNet-OOD: Deciphering Modern Out-of-Distribution Detection Algorithms." arXiv preprint arXiv:2310.01755 (2023).

[f] Hsu, Yen-Chang, et al. "Generalized odin: Detecting out-of-distribution image without learning from out-of-distribution data." Proceedings of the IEEE/CVF conference on computer vision and pattern recognition. 2020.

[g] Tian, Junjiao, et al. "Exploring covariate and concept shift for out-of-distribution detection." NeurIPS 2021 Workshop on Distribution Shifts: Connecting Methods and Applications. 2021.

[h] Yang, Jingkang, et al. "Generalized out-of-distribution detection: A survey." International Journal of Computer Vision (2024): 1-28.

[i] Baek, Eunsu, et al. "Unexplored Faces of Robustness and Out-of-Distribution: Covariate Shifts in Environment and Sensor Domains." Proceedings of the IEEE/CVF Conference on Computer Vision and Pattern Recognition. 2024.

[j] Averly, Reza, and Wei-Lun Chao. "Unified out-of-distribution detection: A model-specific perspective." Proceedings of the IEEE/CVF International Conference on Computer Vision. 2023.

[k] Pagliardini, Matteo, et al. "Agree to disagree: Diversity through disagreement for better transferability." arXiv preprint arXiv:2202.04414 (2022).

[l] Lee, Yoonho, Huaxiu Yao, and Chelsea Finn. "Diversify and disambiguate: Learning from underspecified data." arXiv preprint arXiv:2202.03418 (2022).

[m] Rame, Alexandre, et al. "Diverse weight averaging for out-of-distribution generalization." Advances in Neural Information Processing Systems 35 (2022): 10821-10836.

[n] Chu, Xu, et al. "Dna: Domain generalization with diversified neural averaging." International conference on machine learning. PMLR, 2022.

[o] Lin, Yong, et al. "Spurious feature diversification improves out-of-distribution generalization." arXiv preprint arXiv:2309.17230 (2023).

[p] Xia and Bouganis: On the Usefulness of Deep Ensemble Diversity for Out-of-Distribution Detection https://arxiv.org/abs/2207.07517

[q] Wang, Dequan, et al. "Tent: Fully test-time adaptation by entropy minimization." arXiv preprint arXiv:2006.10726 (2020).

[r] Teney, Damien, Maxime Peyrard, and Ehsan Abbasnejad. "Predicting is not understanding: Recognizing and addressing underspecification in machine learning." European Conference on Computer Vision. Cham: Springer Nature Switzerland, 2022.

[s] Ovadia, Yaniv, et al. "Can you trust your model's uncertainty? evaluating predictive uncertainty under dataset shift." Advances in neural information processing systems 32 (2019).

[t] Breiman, Leo. "Bagging predictors." Machine learning 24 (1996): 123-140.

[u] Teney, Damien, et al. "Id and ood performance are sometimes inversely correlated on real-world datasets." Advances in Neural Information Processing Systems 36 (2024).

---

### Decision · Program_Chairs · 2024-09-25

**Decision:**

Reject

**Comment:**

The submission proposes a method for diversifying ensembles for learning problems that rely on ensembles in their methodology. The submission and the rebuttal were carefully considered, and the AC asked for discussion among the reviewers given the ratings. In the end, the submission didn't generate excitement, and none of the reviewers gave a rating stronger than borderline. After the discussion, the consensus of the reviewers was not to support the submission for acceptance in its current form. Their comments during the discussion are copied below since they are not visible to the authors. A basis for acceptance could not be established.

"I did not find the paper exciting. There is some consensus that the experiments were not comprehensive and the results were not convincing."

"I also did not find the idea to be exciting. I think the premise of diversifying on "OOD" examples within the ID data is interesting, but clearly leads to worse ID performance, so this does not seem very useful in practice. The contributions for speeding up the pairwise disagreement also seems incremental, since it is the exact same formula but sampled stochastically."

"I think the paper is technically solid (i.e. no big flaw) and has made some contribution, but I don't find it particularly interesting or exciting either, so I maintain borderline acceptance.

The main reason why this paper does not quite interest me is probably that it lacks in-depth analyses/discussion. It offers little insight into why the method works---specifically, why would diversifying models' predictions on hard training samples help OOD generalization? Overall, it seems the paper barely deepens our understanding of the subject, so I have reservations about the paper's impact. The authors acknowledged the issue in the rebuttal but the proposed changes are still somewhat vague and therefore insufficient to fully address the issue."

"the authors did a lot of work in the rebuttal to address this, although given the time frame, the new results were sometimes on limited setups. E.g., a main contribution was finding "an OOD subset in the ID dataset", however, simple heuristic augmentations can also introduce OOD examples in a much cheaper way than what they proposed, the authors added these new results in the rebuttal but understandably, on a small scale."